# COVID-19: A Comprehensive Review on Cardiovascular Alterations, Immunity, and Therapeutics in Older Adults

**DOI:** 10.3390/jcm12020488

**Published:** 2023-01-06

**Authors:** José Rivera-Torres, Natalia Girón, Esther San José

**Affiliations:** Department of Health Sciences, Faculty of Biomedical and Health Sciences, Universidad Europea de Madrid, 28670 Villaviciosa de Odón, Spain

**Keywords:** SARS-CoV-2, COVID-19, elderly, cardiovascular diseases, heart failure, cytokine storm, immune response, cytokine antagonist, vaccines

## Abstract

Here, we present a review focusing on three relevant issues related to COVID-19 and its impact in older adults (60 years and older). SARS-CoV-2 infection starts in the respiratory system, but the development of systemic diseases accompanied by severe clinical manifestations has also been reported, with cardiovascular and immune system dysfunction being the major ones. Additionally, the presence of comorbidities and aging represent major risk factors for the severity and poor prognosis of the disease. Since aging-associated decline has been largely related to immune and cardiovascular alterations, we sought to investigate the consequences and the underlying mechanisms of these pathologies to understand the severity of the illness in this population. Understanding the effects of COVID-19 on both systems should translate into comprehensive and improved medical care for elderly COVID-19 patients, preventing cardiovascular as well as immunological alterations in this population. Approved therapies that contribute to the improvement of symptoms and a reduction in mortality, as well as new therapies in development, constitute an approach to managing these disorders. Among them, we describe antivirals, cytokine antagonists, cytokine signaling pathway inhibitors, and vaccines.

## 1. Introduction

Coronaviruses are a large family of enveloped, single positive-stranded RNA viruses that infect both animal species and humans (Figure 1). A novel member of the beta-coronavirus genus was identified in December 2019 in Wuhan (China) as the etiological agent of severe acute respiratory syndrome (SARS-CoV-2) [1]. Rapidly, the virus spread worldwide, and the World Health Organization declared the outbreak as a pandemic on 11 March 2020. Since then, this organization reported on 5 May that between 13.3 million and 16.6 million people had died because of the pandemic.

Increasing age represents one of the major risk factors for the frailty of the human being. This is especially true for the immune and cardiovascular systems, impacting their functions. Furthermore, the elderly population is more susceptible to viral infections, which are accompanied by a worse prognosis [2].

A comprehensive report of more than 72,000 cases in China [3] showed that case fatality rates increased in an age-dependent manner (8.0% in patients aged 70–79 years; 14.8% in patients aged ≥ 80 years vs. less than 1% for those under 50). More importantly, comorbidities appear to significantly impact COVID-19 severity and increase the risk of death [4]. Specifically, associated cardiovascular diseases and hypertension raised case fatality rates to 10.5% and mortality to 6%, respectively, from 0.9% observed for people with no comorbidities [3,5] and largely increased the risk and severity of COVID-19 infection [6].

Furthermore, the hyperinflammatory response, triggered by cytokine and chemokines, observed in COVID-19 patients seems to play a key role in the pathogenesis by either revealing an undiagnosed disease or provoking direct tissue damage. Therefore, understanding the damage caused by SARS-CoV-2 to the cardiovascular system as well as the impairment of the immune system, together with the underlying mechanisms, is of the greatest importance for the benefit of patients. Therapies to treat the severe acute respiratory syndrome caused by COVID-19 (approved or under clinical investigation) might prevent cardiovascular alterations as well as the dysregulation of the immune system.

Within this review, we summarize the current knowledge supported by clinical findings in both the cardiovascular and immune systems. Finally, a summary of current therapies related to the former topics is included.

## 2. SARS-CoV-2 and Cardiovascular Alterations

Patients suffering from COVID-19 typically show respiratory distress accompanied by, among other signs and symptoms, an excessive inflammatory response and cardiovascular complications leading to multiorgan damage, which constitutes a key threat. Interestingly, cardiovascular alterations have been reported with no signs of respiratory disease. In addition, respiratory symptoms are worse in COVID-19 patients in the setting of chronic cardiac disturbances.

Accordingly, SARS-CoV-2 has been found to interact with and affect the cardiovascular system by latching onto host cells via the angiotensin-converting enzyme 2 (ACE2) receptor, which is expressed not only in the lungs but also in other tissues, including the heart and endothelium [7] (Figure 1). This interaction leads to microvascular dysfunction [8], among other alterations. Similarly, TMPRSS2, cathepsin L, and furin, accessory proteins involved in the priming process that the viral S protein undergoes to facilitate SARS-CoV-2 binding to its cognate ACE2 receptor, are also expressed in the endothelium of coronary arteries, as well as in cardiomyocytes [9,10].

Among cardiovascular alterations, COVID-19 patients commonly display myocardial injury, accompanied or not by clinical manifestations (Figure 1). Although hypoxia [11], the weakening of the left ventricle (Takotsubo syndrome) [12], endothelial damage and vascular dysfunction [13], myocarditis [14], and a systemic inflammatory response known as a cytokine storm [15] underlie the disparate effects of SARS-CoV-2 infection on patients with and without cardiovascular comorbidities, their mechanisms and contributions remain incompletely understood.

A broad body of evidence suggests that patients with severe symptoms often have complications leading to cardiomyocyte death, thus involving myocardial damage. A comprehensive meta-analysis revealed that the incidence of myocardial damage rose to 30% in patients over 60 years old, two-fold more than in younger patients [16]. The assessment of injury is primarily accomplished by determining the levels of the bona fide circulating biomarker troponin and, more precisely, the high-sensitivity cardiac troponin I (hs-cTnI) isoform [17].

Between 10 and 30% of patients requiring treatment at healthcare units show cardiac troponin elevation. It is widely accepted that the higher the troponin level, the worse the prognosis. Thus, elevated troponin levels are highly associated with increased mortality [18,19], although patients do not display suggestive acute coronary syndrome. Similarly, the elevation of N-terminal pro-B-type natriuretic peptide (NT-pro) BNP, creatine phosphokinase (CK)-MB, and myoglobin (MYO) cardiac injury markers above established cutoffs, a measure of cardiomyocyte death, was associated with an increased risk of death [20]. However, these markers are not capable of distinguishing between diverse causes accounting for myocardial damage, which encompasses, among others, myocarditis, cardiomyopathy, and myocardial infarction.

Since the identification of the first small (17–25 nucleotides) non-coding RNAs in 1993, microRNAs (miRNAs) have become a valuable tool in the research field, being involved in diverse cardiovascular alterations, including cardiac hypertrophy, ischemic heart disease, and heart failure, among others. Furthermore, since their identification in the circulatory system, miRNAs have been considered not only a potential biomarker but also a plausible therapy against hepatitis C virus infection (Miravirsen). 

Importantly, a recent review that focused on the role of miRNAs in cardiovascular complications associated with COVID-19 shed light on their potential as valuable biomarkers and predictors of both cardiac and vascular damage upon SARS-CoV-2 infection. Additionally, the authors discuss their utility as potential therapeutic targets for COVID-19 patients [21].

**Myocarditis and inflammatory cardiomyopathy**. Since the first report detecting the SARS-CoV-2 genome in heart biopsies from patients diagnosed with myocarditis or unexplained heart failure [22], many case reports studies have identified this alteration in COVID-19 patients [23,24,25,26,27,28,29]. However, reports rarely demonstrate this clinical condition [22,30,31] independently of its origin. 

Among the conditions underlying this injury, SARS-CoV-2 viral infection has been postulated, but confirmation remains on hold [22,32,33].

Proof of concept that SARS-CoV-2 triggers viral myocarditis requires not only excluding the presence of well-known cardiotropic viruses [34] that display inflammatory features like SARS-CoV-2 but also its specific identification in and/or isolation from cardiac cells. Additionally, demonstrating the presence of inflammatory cells that are directly linked to cell death and clearly distinguished from ischemic damage is required [35].

A German study carried out in 39 autopsy cases of COVID-19, where the median age was 85 years, documented SARS-CoV-2 presence within the heart tissue in 61.5% of cases [36]. All the analyzed tissues displaying high copy numbers demonstrated increased levels of proinflammatory cytokines (TNF-α; IFN-γ; CCL5; and IL-6, -8, and -18) versus non-heart-infected patients. However, the comparison revealed that infection was not associated with an influx of inflammatory cells, necessitating further investigations to study the long-term consequences. Direct, and not viral, immune-mediated myocarditis has been suggested, but more data are required to strongly support this hypothesis. The absence of studies demonstrating proof of cardiac inflammation and firmly establishing the lack of viral presence by molecular techniques within the same biopsy specimens represents a challenging task since, in such scenarios, it is not possible to exclude alternate causes of the inflammation [32].

Potential associations between SARS-CoV-2 vaccines, both mRNA (BNT162b2 and 1273, manufactured by Pfizer and Moderna, respectively) and recombinant nanoparticles (NVX-CoV2373; Novavax), and myocarditis and pericarditis events have been reported. From mRNA formulations, signs have been reported after a week following the second booster. A recent document from the FDA Advisory Board (FDA Briefing Document Novavax COVID-19 Vaccine) reported multiple events temporally related to NVX-CoV2373 administration in two different studies evaluating efficacy, safety, and immunogenicity (Study 2019nCoV-301) and an ongoing phase 3 trial (Study 2019nCoV-302). Myocarditis was mainly observed, but not uniquely, in the group of young male adults (18 to ≤64 years old), whose symptoms resolved in the short term without sequelae. This unexpected pattern of adverse events necessitates a larger randomized trial, mainly over the long term, to assess safety in this more vulnerable elderly population.

Plausible explanations underlying these effects are the cross-reactivity of antibodies raised against the viral spike glycoprotein with contractile proteins in the heart [37], as well as the ability of the mentioned antibodies, in association with several factors, to trigger myocardial damage [38]. In contrast, these symptoms have not been reported so far upon inoculation with the replication-deficient human type 26 Adenovirus-based vaccine (Ad26.COV2.S; Johnson & Johnson).

**Heart failure**. According to the results observed in two independent retrospective studies carried out on COVID-19 patients in China, heart failure was the most common complication not involving the respiratory system [19,39]. Importantly, this manifestation was associated with higher mortality during hospitalization when compared to survivors [19].

Recently, a follow-up study encompassing a large cohort of COVID-19 patients in the US [40] shed light on long-term cardiovascular outcomes. The authors estimated the cardiovascular risks beyond the first month after infection and demonstrated an increased risk of diverse cardiovascular alterations, independent of whether patients underwent hospitalization or not. The increased risk was most pronounced for heart failure, followed by atrial fibrillation, regardless of the presence of other cardiovascular risk factors.

Although the authors [40] suggest that SARS-CoV-2 infection may also lead to the appearance of a novel cardiovascular disease, whether heart failure arises because of a worsened pre-existing pathology or due to a new cardiac dysfunction, as shown above, remains unclear, necessitating further efforts to address healthcare strategies that will be required to treat affected people.

**Cardiogenic shock**. This represents a common cause of mortality caused by the severe impairment of myocardial outcomes, leading to hypoxia, hypoperfusion, and diminished cardiac output [41]. Hypoxia, an excessive inflammatory response, and vasodilatory shock as the SARS-CoV-2 infection progresses might underlie this fatal condition. Outcomes are aggravated by pre-existing cardiovascular dysfunction.

A case report study [42] demonstrated that elevated levels of SARS-CoV-2 antibodies might play a pathogenic role in the hyperinflammatory response. This viral–host interplay seems to facilitate viral entry mediated by an antibody-dependent enhancement mechanism, as we explain later. Signs and symptoms were resolved quickly following treatment with a moderate dose of steroids in this population.

More importantly, older patients with cardiogenic shock display lower short-term survival, along with a higher risk of death [43].

**Cardiac arrhythmias**. A multicenter study performed in hospitalized COVID-19 patients observed a variety of arrhythmic manifestations, ranging from benign to potentially life-threatening, with sinus tachycardia (rate >100 beats per minute) being the commonest [44]. Alterations in the heart’s rhythm have been mainly reported in COVID-19 patients who were transferred to the intensive care unit, with 44.4% compared to 16.7% of patients not requiring intensive care treatment [1]. More recently, a meta-analysis encompassing a reduced number of studies on bradycardia and mortality in COVID-19 patients showed that bradycardia was not statistically associated with mortality in COVID-19 patients [45]. 

In the context of a viral infection, different factors have been linked to this outcome. The upregulation of the cognate ACE2 viral receptor in the heart tissue may facilitate viral entry into the conduction system, leading to arrhythmia. Medications prolonging the QT segment [46,47], hypoxia and electrolyte disturbances, a proinflammatory state, abnormal metabolism [48], and cardiovascular comorbidities or complications in the setting of a viral infection may also more easily trigger cardiac dysrhythmia. 

**Acute myocardial infarction**. Fortunately, most COVID-19 patients display mild symptoms or are asymptomatic. However, more severe clinical manifestations, including pneumonia and acute respiratory distress, are augmented in older patients, diagnosed or not, with heart and/or cardiovascular affectation. 

Elevated troponin levels are common in COVID-19 patients hospitalized because of stress cardiomyopathy, ischemia, or the systemic release of inflammatory cytokines. A minority of them show symptoms and signs suggestive of acute coronary syndrome. Additionally, cardiovascular-associated diseases, obesity, and diabetes increased the risk of a poor prognosis, with myocardial injury patients being the ones with poorer prognoses.

According to a comprehensive review [17], the frequency of myocardial injury varies widely, ranging from 7 to 28%, and there is a strong correlation between myocardial injury and illness severity. 

Although the evidence that SARS-CoV-2 viral infection increases the risk of acute myocardial infarction is mainly based on case reports, a Danish study encompassing all patients hospitalized and diagnosed with COVID-19 in the country [49] paved the way for the assessment of this association. Signs and symptoms were more pronounced in patients suffering from prior cardiovascular disease and in the elderly. 

**Stress cardiomyopathy**. This condition, also known as Takotsubo syndrome, develops upon intense emotional or physical stress, leading to acute but reversible left ventricular dysfunction. Affected people display similar symptoms (altered echo- and electrocardiogram profiles) to those observed upon myocardial infarction but with no obstructive coronary artery disease.

Excessive cytokine release (hereafter, cytokine storm) has been linked to a catecholamine surge, which enhances inflammatory injury [50]; thus, it might represent a pathophysiological mechanism of the development of or predisposition to Takotsubo syndrome in COVID-19 patients [12]. The first systematic review [51] in COVID-19 patients (mean age 70.8) concluded that inflammation (cytokine storm and cardiovascular comorbidities) and physical (intubation) and/or emotional stresses (social isolation and associated anxiety) underlie the syndrome, which was predominant in older women. The absence of important coronary lesions almost ruled out the role of acute coronary syndrome as the culprit of Takotsubo syndrome. This study encompassed a reduced number of patients, and some of them have been recommended to be excluded [52] from the systematic review. More studies are required to shed light on this topic.

**Coagulation abnormalities**. SARS-CoV-2-infected patients demonstrate a wide range of coagulation abnormalities accompanied by increased levels of circulating prothrombotic factors, such as factor VIII, fibrinogen, and prothrombotic microparticles [53,54].

Similarly, elevated neutrophil counts [55] and D-dimer levels, a degradation product of cross-linked fibrin indicative of augmented thrombin generation and fibrin dissolution, have been reported [19,56]. Altogether, coagulation abnormalities can lead to thromboembolic events, a common complication in critically ill patients with COVID-19, either in veins [57] or in arteries [58].

The virus’ ability to directly infect endothelial cells underlies cell damage and, thus, can account for the vast number of effects reported in COVID-19 patients, such as micro-vascular inflammation, endothelial exocytosis, and endotheliitis (reviewed in [59]).

The direct and specific activation of the alternative complement pathway mediated by the viral S protein has been reported to contribute to many of the clinical manifestations, such as microangiopathy, thrombocytopenia, and thrombophilia, among others [60]. To be precise, COVID-19 patients demonstrate elevated markers of complement activation in either the blood [61], kidney [62], or skin [63]. Elevated levels of a specific complement activation marker show a direct relationship with disease severity. Accordingly, the administration of eculizumab, a monoclonal antibody raised against the C5 complement factor, dramatically diminished both elevated D-dimer levels as well as neutrophil counts and led to the remission of symptoms [63].

A hyperinflammatory state upon SARS-CoV-2 infection might also account for coagulation alterations since cytokine IL-6 acts as a transcriptional regulator of acute-phase Coagulation Factor VIII [64].

Since the main coagulation abnormalities in COVID-19 patients suggested a hypercoagulable state, some retrospective studies sought to demonstrate that systemic anticoagulation was associated with prolonged survival in hospitalized patients independently of the type of treatment, either by using a systemic anticoagulation treatment [65] or by using low-molecular-weight heparin [66]. Both reports concluded that the median survival time of patients who were treated with anticoagulation was longer than in those who were not treated.

## 3. Immunosenescence and Inflammaging in the Older Adults 

It is well known how the immune system can recognize foreign pathogens and become activated, with several consequences: the fighting of the pathogen, the development of memory, and occasionally, the development of an inflammatory pathology upon the dysregulation of the immune system itself [39]. In general, the response mediated by the immune system consists of the mobilization of white blood cells. In the innate immune response, macrophages, dendritic cells, and cytokines, among others, play a central role that permits the activation of the adaptive immune system by relying on T and B cells, key protagonists for the development of immune memory.

In older adults, all of these processes are affected, because aging leads to both physical and physiological deterioration. The production of leukocytes in the bone marrow is strongly affected by age, giving rise to a decline in the activation of the innate and adaptive systems. In consequence, the individual is led to a state rendering them prone to more easily get infected and to develop cancer or even autoimmune diseases [67,68]. A general decrease in the cell cycle activity of hematopoietic stem cells (HSCs) has been described in studies on aged mice [69].

Recently, a new term has been proposed for all of these events, immunosenescence, which refers to the age-related remodeling of the immune system, giving rise to an exacerbated inflammatory response and increasing susceptibility to respiratory infections [70]. This leads to an alteration in intercellular communication giving rise to a latent proinflammatory state, named “inflamm-aging”, with elevated levels of inflammatory mediators in the serum, such as IL-6, IL-1RA, TNF-α, IL-1, and C-reactive protein (CRP), which are responsible for a delay in or even the absence of the activation of the immune system [71,72]. Other causes of this inflammaging are the exacerbated activation of the NF-kB transcription factor as well as a deficiency in the autophagic response necessary for promoting cellular survival by maintaining cellular energy levels in case of starvation [73]. Furthermore, senescent cells undergo telomeric disorders and oxidative stress, giving rise to the augmentation of cytokines, chemokines, lipids, and growth factors [74,75] (Figure 2A).

Finally, throughout the life of the individual, the thymus loses its ability to generate new clones of T lymphocytes, leaving the elderly population in a state without virgin T cells and, therefore, in an unprotected condition against foreign agents [76].


**
*COVID-19 and the Immune System in Older Adults*
**


Several risk factors, such as age, male sex, obesity, smoking, hypertension, type two diabetes mellitus, etc., have been reported to influence the development of severe COVID-19 [3,19,77]. Age is considered the most relevant one, as we noted before, not only for the development of COVID-19 but also for its adverse health outcomes [39]. Studies in aged Syrian hamsters infected with SARS-CoV-2 support this insight [78]. Therefore, the immune system dysregulation that occurs during the aging process can contribute to the pathogenesis of the disease.

In 2020, at the beginning of the pandemic, SARS-CoV-2 was a new pathogen, and the whole population was unprotected because there was no herd immunity. The hypothesis that the elderly population might be protected by immunological memory cross-reactivity with other coronaviruses has been tested, but more studies should have been carried out in larger cohorts [79,80,81,82]. Nevertheless, the clinical outcomes reveal the opposite to be true, and as we mentioned above, CFR increases with age, as has been shown in many countries [39].

Zhang et al. performed a complete single-cell analysis of human circulating immune senescent cells and immune cells, comparing young and aged COVID-19 patients at the transcriptomic and protein levels, and discovered changes at the cellular and gene levels involved in both inflammation and cellular senescence, giving rise to the vulnerability shown by the older adults [83].

By exploring the molecular and cellular mechanisms that this infection has in the elderly population, we can observe some important points regarding the immune system. In this review, we will summarize some of the more important ones, considering that there are many more. 


**
*Innate Response and Inflammation in COVID-19 in the Older Adults*
**


The severity of the disease is marked not only by the virus itself but also by an aberrant immune response. Three different phases have been described clinically. First, there is an immune response against the virus, triggering an inflammatory response. Symptoms include fever, dry cough, diarrhea, and headache, which could be complicated in a second phase by dyspnea and hypoxia. In the worst cases, the third phase, acute respiratory distress syndrome (ARDS), develops, requiring intubation and ventilator support and occasionally provoking multiorgan failure and death [84]. It is well described that the inflammatory response is due to excessive immune activation and the production of proinflammatory cytokines, leading to a cytokine storm. This process takes place in the lungs and is one of the key points in the disease associated with SARS-CoV-2 in older adults.

Monocytes. On the one hand, monocytes migrate to the lungs and secrete proinflammatory cytokines, and on the other hand, alveolar macrophages in the elderly can have an impaired response, increasing cytokine production and causing a deficiency in tissue repair [85,86]. Furthermore, most people (more than 82%) who experience this cytokine storm are over the age of 60 [87]. The most common cytokines secreted include IL-6, IL-1, and GM-CSF by monocytes. As discussed later, several therapies mostly blocking IL-6 and IL-1 have been proposed with relative success, together with the inhibition of JAK-STAT kinases, essential for signaling inside cells once the cytokines have contacted their receptors. 

Antigen-presenting cells. It is important to address the fact that antigen-presenting cells in elderly people (mostly macrophages and dendritic cells) display a decrease in CD80 and MHC class II molecule expression, accompanied by mitochondrial dysfunction indicative of a likely deficient adaptive immune response [88,89,90].

Neutrophils. On the other hand, another component of the innate response, neutrophil cells, have been found in higher amounts in the peripheral blood of SARS-CoV-2-infected individuals with advanced age and in autopsied COVID-19 patients, leading to tissue damage [91,92]. These cells in older adults have a deficiency in phagocytosis with lower microbicidal activity [93]. It has been proposed that neutrophils could promote coagulopathy, termed immunothrombosis, due to neutrophil infiltration and the formation of neutrophil extracellular traps (NETs) [94]. These NETs seem to be responsible for disease progression in many tissues, such as the lungs, heart, or even kidneys [95]. 

NK cells, important for the first viral clearance before the adaptive response takes place, are also affected in elderly people. There are two main groups of NK cells: one responsible for virus fighting and the other having an immunomodulatory function. In older adult people, there is a decrease in IFN-γ production that mediates a drop in the CD56bright population, which could explain the vulnerability of elderly people to microorganism infections due to the impairment of cytokine release that mediates this cell subtype [96]. (For a summary of the impact of COVID-19 on the innate immune response in elderly people, see Figure 2B.)

Complement system. Another main component of the innate immune response is the complement system. It has been described that SARS-CoV-2 mediates its activation via the lectin pathway. The serum protein MBL (mannose-binding lectin) recognizes mannose residues in the ACE2 protein, responsible for virus entry. A reduction in virus replication due to the activation of the complement system has been described, and it has even been suggested that MBL binding to ACE2 could inhibit virus entry [97,98]. In the case of elderly people, Tomaiuolo R et al. revealed that the MBL serum concentration was significantly lower compared to the general population, indicating that, likely, the complement system is not appropriately activated [99] (Figure 2A).

Alternatively, to the activation of the complement system, another host defense mechanism is the triggering of the coagulation cascade. Patients suffering from a cytokine storm will be at risk of intravascular coagulation and endothelial injury, as mentioned above, giving rise to thromboinflammation [100].


**
*Adaptive Response in COVID-19 Older Adults*
**


T cells and cell-mediated immunity. Commonly having a good prognosis, COVID-19 patients have a high T lymphocyte (CD4 and CD8 T cells) count due to the clonal expansion that takes place after encountering an antigen. CD8 T cells are important for the clearance of the virus in a very specific way, whereas CD4 T cells are key to orchestrating an effective and strong response, triggering the activation of other cell types, such as B cells, CD8 T cells and macrophages. 

Fewer T lymphocytes in severe and aged COVID-19 patients have been described compared with young people, with the CD8 cell population being the one most affected [101]. This reduction leads to a deficient cell-mediated response, and thus, it has been proposed as a biomarker of a poor prognosis [102]. The reasons underlying this lymphopenia should be figured out. On the other hand, the impact that many viruses, including SARS-CoV-2, has on the impairment of IFN type 1 secretion (viral immune evasion) is well known, which could explain the poor response mediated by CD8 T cells [103]. However, it cannot explain the lymphopenia itself. 

As mentioned before, due to thymus atrophy in elderly people, there is a reduction in naïve T cells, which compromises the T-cell receptor (TCR) repertoire present in this group (Figure 2A). There is an increase in memory T cells that will not be able to fight against new pathogens [104]. 

Several studies reveal that T cells in elderly people express PD1+ and Tim3+, markers of exhausted cells with a limited capacity to induce an efficient response [105], and they also have limited IL-2 secretion, which is necessary for T-cell expansion [106].

A successful immune response entails a good balance between Th1 and Th2 (subtypes of CD4 T cells). Importantly, there is an increase in cytokines that mediate the polarization to Th2, accompanied by a marked reduction in IFN-γ secretion, in old people infected with SARS-CoV-2, leading to the inefficient activation of CD8 T cells [107]. 

Another important subtype of helper T cells is the Th17 population. It has been stated that there is polarization toward this cell type due to an increased amount of IL-6 in the environment. These effects lead to the blockage of Th1 polarization. IL-7 secretion by Th17 also induces the activation of phagocytes, giving rise, therefore, to the cytokine storm [108]. (For a summary of the impact of COVID-19 on the adaptive response in elderly people, see Figure 2B.)

Finally, it is important to highlight the correlation that could exist between specific genes and COVID-19 severity. Several authors have demonstrated that common and rare genetic variants that influence the IFN signaling pathways affect the tendency to develop severe COVID-19. Although the prevalence of the impact is 1–5% in young people (≤60 years of age), this fact should be taken into consideration in elderly people [109,110].

B cells and the humoral response. Similar to the case for T cells, the older adults has a higher number of memory B cells with a limited B-cell receptor (BCR) repertoire, which translates to an inefficient humoral response mediated by antibodies. The normal sequence of antibody production involves the secretion of specific IgM and IgG/IgA against viral N and S proteins, and several studies demonstrate the presence of such antibodies in severe cases as well as among mild cases [111].

It has been described that aged mice infected with influenza A show a reduction in IgG production, which is required for a sustained immune response, compared with IgM, which was revealed to be unaffected [112]. Other important requirements for the activation of B cells and the secretion of high-affinity antibodies are affected, as has been demonstrated in both aged animal models and older humans. The CD40L-CD40 interaction between B and CD4 T cells, indispensable for isotype changes, is very weak [113], and there is a deficiency in switching on the activation-induced cytidine deaminase (AID) enzyme necessary for the process of antibody improvement, known as somatic hypermutation [114]. Both events are essential for the suitable activity of antibodies. Taking these data together and summarizing, the humoral response in older individuals is affected, and furthermore, it is aggravated in COVID-19 patients with comorbidities, where lower levels of specific antibodies are detected compared with only-SARS-CoV-2-infected subjects [115].

An antibody-dependent enhancement (ADE) mechanism involving the binding of non-neutralizing antibodies to FcγR, present in many cell types, represents another major concern. This effect could favor the replication of SARS-CoV-2 in lung tissues, mostly in aged individuals [116,117]. It is important to address that neutralizing antibodies present in vaccinated patients or in those who receive plasma from recovered subjects do not activate the ADE mechanism [118,119].

## 4. Therapeutics against SARS-CoV-2

SARS-CoV-2 infection causes the impairment of the immune system, together with cardiovascular damage, as noted above; therefore, therapies to block the replication or entry of the virus, as well as the induced cytokine release, represent a strategy to prevent immune and cardiovascular complications.

Currently, many approaches to handling and preventing the infection are available or under investigation. Among them, corticosteroids, a group of anti-inflammatory drugs, constitute the first line of defense as immunomodulators to reduce excessive cytokine release. However, their immunosuppressive ability accounts for the reduction in lymphocyte counts, consequently increasing the risk of superinfections, mainly in the geriatric population [120]. Thus, clinical guidelines recommend their use (mainly dexamethasone) for moderately and critically ill patients requiring low-flow supplemental oxygen and ventilatory support [121].

Susceptibility to infection and disease severity may be increased using angiotensin type-1 receptor blockers (ARBs) and ACE inhibitors (ACEIs), since both increase the viral receptor (ACE2) levels, and thus, targeted therapies to block the ACE2 protein are being considered. However, preclinical studies indicate that ACEIs and ARBs can be beneficial for COVID-19 outcomes by reducing the formation (ACEIs) or blocking the action (ARBs) of angiotensin II, avoiding the risk of cardiovascular damage. This fact might need to be considered in some cases for discontinuing treatment [122,123].

Accordingly, many studies have evaluated the effects of these types of drugs in COVID-19 patients, demonstrating a decrease in morbidity in patients who received treatments with ACEIs or ARBs [124]. In fact, patients treated with these drugs showed a decrease in the risk of severe disease development compared to patients without treatment [125]. Other studies have evaluated the effect of the withdrawal of these types of drugs in COVID-19 patients and concluded that the discontinuation of this treatment resulted in an increase in mortality [126] or that there was no significant difference in the mean number of days alive and out of the hospital for patients hospitalized with mild to moderate COVID-19 who were taking ACEIs or ARBs before hospital admission [127]. 

The use of known or novel antiviral drugs constitutes an alternative approach. These drugs may act at different viral cycle steps. Next, we briefly review those that are mostly involved in viral entry or genome replication.

Hydroxychloroquine (HCQ), a drug derived from the antimalarial drug Chloroquine and used to treat conditions such as rheumatoid arthritis, was proposed at the start of the pandemic because of its ability to prevent lysosomal acidification, a likely path for SARS-CoV-2 cell entry. However, clinical trials assessing the efficacy of HCQ vs. remdesivir, lopinavir, or IFNβ-1a treatments concluded that HCQ had no impact on survival or the need for mechanical ventilation [128] and also demonstrated cardiotoxicity after its short-term use [129].

Camostat mesylate, a serine protease inhibitor that targets the TMPRSS2 protease that primes the viral spike protein, might be a potential antiviral drug. Preclinical studies have shown that camostat mesylate inhibits viral-envelope-to-cell-membrane fusion and thus viral entry and uncoating [130]. Clinical trials (phases II and III) are ongoing to assess its safety and efficacy (NCT04455815, NCT04583592, and NCT04608266, among others). 

RNA-dependent RNA-polymerase inhibitors or drugs blocking viral replication are widely used as antiviral compounds. Favipiravir or remdesivir might represent an effective approach. Specifically, remdesivir was approved by the FDA and EMA to treat patients with pneumonia who may or may not require supplemental oxygen but are at increased risk of developing severe COVID-19. Similarly, favipiravir showed an improvement in clinical symptoms [131]. Late last year, molnupiravir was also authorized by the FDA for the treatment of mild-to-moderate COVID-19 in adults. Alternate antiviral drugs such as Umifenovir (an anti-influenza drug that blocks viral entry mediated by endocytosis) exhibited statistically significant efficacy for mild/asymptomatic patients in a phase III clinical trial [132]. However more clinical studies are needed to evaluate the efficacy and safety of these antivirals.

Next, we summarize current treatments capable of managing the cytokine storm or its downstream signaling cascade and preventing COVID-19 infection. Finally, we provide the latest knowledge on the development of vaccines.

**Antagonists of IL-6.** Many compounds have been raised against the inflammatory cytokine IL-6 and its membrane-bound receptor (IL-6R). Tocilizumab, Satralizumab, Siltuximab, Clazakizumab and Levilimab are antagonist drugs that block the IL-6 signaling pathway activated upon SARS-CoV-2 infection.

The monoclonal antibody tocilizumab (TCZ) has been approved for the treatment of critically ill COVID-19 patients who are receiving corticosteroids and require supplemental oxygen, non-invasive or invasive mechanical ventilation or extracorporeal membrane oxygenation. The RECOVERY study assessed its efficacy in COVID-19 patients versus placebo, presenting a significant decrease in 28-day mortality in hospitalized patients with severe COVID-19 [133]. Salvarini et al. assessed its efficacy in a multicenter, open-label randomized clinical trial and demonstrated a decrease in mortality risk in critical patients older than 60 years old [134].

Further studies confirmed the efficacy of TCZ combined with other IL-6R antagonists, such as Sarilumab [133] or Siltuximab [135], exhibiting reduced 28-day mortality compared to the standard of care at intensive care units.

Sirukumab, Siltuximab, and Clazakizumab, monoclonal anti-IL-6 antibodies, are being studied as monotherapy strategies in clinical trials (NCT04380961, NCT04322188, and NCT04381052, respectively) to reduce the hyperinflammatory state in older patients with severe COVID-19. Additionally, a combined treatment of Siltuximab and corticosteroids is also under assessment (NCT04486521).

Levilimab, an IL-6R inhibitor, in a combination with the standard of care in severe patients not requiring ventilation resulted in an improvement in the sustained clinical rate [136].

**Antagonist of IL-1.** The utilization of the IL-1 antagonist Canakinumab has been proposed as an effective treatment for moderate COVID-19 patients. A prospective case–control study revealed that patients (median age 70 years old) with moderate pneumonia rapidly recovered normal oxygen levels upon Canakinumab treatment. This effect was accompanied by a reduction in lung tissue damage and inflammation markers, which were not observed in the cohort receiving the standard of care, leading to earlier hospital discharge and better prognosis [137]. Conversely, a randomized clinical trial including 454 hospitalized patients with severe COVID-19 showed that Canakinumab treatment did not significantly increase (88.8% vs. 85.7% in the placebo group) the likelihood of survival without mechanical ventilation [138].

Anakinra has been approved to treat COVID-19 in adults who have pneumonia requiring supplemental oxygen (low- or high-flow oxygen) and who are at risk of developing severe respiratory failure. In fact, this drug showed that the early initiation of treatment, combined with the standard of care, reduced the risk of worse clinical outcomes at day 28 and mortality in patients hospitalized with moderate and severe COVID-19 [139].

**Inhibitors of the JAK-STAT pathway.** Among the inhibitors that block this pathway, diminishing the levels of inflammatory cytokines, we reviewed baricitinib, ruxolitinib, and tofacitinib.

Baricitinib was approved by the FDA in May 2022 for the treatment of COVID-19 in hospitalized adults requiring supplemental oxygen, non-invasive or invasive mechanical ventilation, or extracorporeal membrane oxygenation. The World Health Organization recommended its use for the treatment of COVID-19 patients because of its ability to reduce the risk of progressive disease and death [140]. Importantly, a study in elderly people (older than 70 years) demonstrated a reduction (18.5%) in the absolute mortality risk in the studied cohort [141].

Furthermore, baricitinib efficacy has been assessed in combination with remdesivir, exhibiting an improvement in efficacy compared to remdesivir monotherapy. The combined treatment reduced the recovery time, accelerating the clinical status of patients. This fact was especially important for subjects receiving supplemental oxygen [142]. The assessment of efficacy for this combination compared to remdesivir plus tocilizumab treatment is under study in critically ill patients (NCT04693026). 

A clinical trial with tofacitinib, another JAK inhibitor, encompassing 289 patients hospitalized with COVID-19 pneumonia lowered the risk of death or respiratory failure through day 28 compared to the placebo group [143]. An open-label randomized control study assessing the efficacy of tofacitinib in addition to the standard of care in hospitalized adults with mild to moderate COVID-19 pneumonia revealed a greater relative reduction in the levels of important markers of inflammation compared to the control group, although there were no differences in hospital discharge or oxygen requirement [144].

Ruxolitinib was investigated in a prospective study performed on patients with severe respiratory failure or high-risk clinical conditions for the management of associated inflammation. In this study, ruxolitinib resolved the hyperinflammatory state in 55% of the patients, regardless of previous steroid or tocilizumab therapy. Unfortunately, few patients demonstrated severe evolution despite ruxolitinib therapy [145]. Another study in 146 hospitalized patients with COVID-19 requiring oxygen support compared ruxolitinib and dexamethasone and reported similar mortality rates [146]. 

Last year, a French pilot study assessed the efficacy of ruxolitinib associated with anakinra for reducing inappropriate systemic inflammatory responses. The preliminary report suggests that the combined treatment may be effective in rescuing critical COVID-19 patients, a population with a high mortality rate [147]. 

Finally, we briefly describe the latest advances regarding the development of different types of vaccines employed as preventive therapies against COVID-19. Among them, we highlight mRNA vaccines (BNT16b2, mRNA1273, and CVnCoV), viral-vector-based vaccines (AZD1222, Ad5-nCoV, and Ad26.COV2.S), inactivated virus vaccines (CoronaVac, BBIBP-CorV, Covaxin, QazVac, KoviVac, and COVIran Barekat), and protein-based vaccines (EpiVacCorona, ZF2001, Abdala).

The main severe adverse cardiovascular events reported in 2021 were very rare after vaccine administration, among which we can underline myocarditis (6 cases per million) and thrombosis with thrombocytopenia syndrome (3 cases per million) for mRNA vaccines and the Janssen vaccine, respectively [148].

A French study assessed the short-term risk of cardiovascular adverse events following the administration of the BNT162b2 mRNA vaccine in people aged 75 years or older, and no increase in the incidence of acute myocardial infarction, stroke, or pulmonary embolism was detected 14 days following each BNT162b2 mRNA vaccine [149]. 

Approved COVID-19 vaccines delivered intramuscularly elicit antibody-mediated and cell-mediated immunity to prevent viral replication and to provide resistance against the development of COVID-19. However, intramuscular (IM) vaccines induce systemic immune responses without generating mucosal protection. 

Recently, two COVID-19 vaccines (viral vector vaccines) that are administrated through the nose or mouth have been approved for use in China and India. China’s inhaled vaccine is delivered through the nose and mouth as an aerosolized mist, and India’s is administered as drops in the nose. Currently, intranasal (IN) vaccines against SARS-CoV-2 are under investigation (viral vector vaccines, recombinant subunit vaccines, and live attenuated vaccines) [150]. Intranasal administration might prevent COVID-19 infection by inducing the secretory IgA antibody response in the nasal compartment. Hence, the administration of an IM dosage followed by an IN vaccination might lead to a robust immune response [151]. 

At the time of this review, 10 clinical trials are ongoing to evaluate the efficacy and safety of 8 intranasal vaccines and 2 intranasal combined with intramuscular vaccines in adults (Table 1). Most of them are phase I trials to assess non-replicating viral vector vaccines (CVXGA1, Gam-COVID-Vac, AdCOVID, NDV, SC-Ad6-1, and NDV-HXP-S) and replicating viral vector vaccines (MV-014-212 and BBV154) in healthy adults. 

A phase II study (NCT05205746) will evaluate the immunogenicity and safety of the NVD vaccine with the administration of a single dose by two different administration routes (IM or IN), and a phase III study (NCT00522335) will recruit participants to receive three consecutive lots of the BBV154 vaccine via the intranasal route and participants to receive the COVAXIN vaccine via the intramuscular route. 

## 5. Conclusions

Older adults (age 60 years and older) commonly display cardiovascular complications and associated comorbidities. Myocardial injury, the most common complication, can be explained by, or be associated with, myocarditis, stress cardiomyopathy, and myocardial infarction. Specifically, further research is needed to clearly establish the relationship between myocardial injury and viral infection. Furthermore, its contribution to the inflammatory state compared to those caused by other respiratory viruses will help to understand the differential symptoms and severity observed. Similarly, the contribution of the hyperproduction of inflammatory cytokines, which leads, among others, to an excessive coagulable state, needs to be addressed. 

Additionally, impairments in the activation of the innate/adaptive immune response due to age and/or virus-related factors have been described. The immune response impairment that takes place physiologically in older adults leaves them in a state with a reduced immune response against pathogens and, in particular, SARS-CoV-2 infection. Studies exploring why there is polarization toward a humoral response, therefore preventing the effective elimination of the virus by the cytotoxic response, should be studied more in depth, as well as the role that macrophages play in the inflammatory response, the cytokine storm and tissue damage. 

Since more and more evidence of the long-term effects of COVID-19 is appearing, it is essential to know whether vaccines will help to prevent the cardiac complications reported above (heart failure, arrhythmias, etc.), as well as the impact that they will have on an effective immune response orchestrating macrophages, T and B lymphocytes, and other immune cells, toward an effective and long-lasting response over time.

Ongoing clinical trials will assess the safety and efficacy of novel therapeutical strategies capable of preventing and reducing the high mortality rate (14.8% in patients aged ≥ 80 years) [3].

Therapies to block the replication or viral entry, as well as prevent the cytokine storm or its downstream signaling cascade, might reduce the cardiovascular and immune alterations caused by SARS-CoV-2 in older adults. In fact, ongoing clinical trials will assess the safety and efficacy of these therapeutical strategies. On the other hand, the development of intranasal vaccines along with vaccines delivered intramuscularly will improve the systemic immune response by generating mucosal protection against SARS-CoV-2. 

## Figures and Tables

**Figure 1 jcm-12-00488-f001:**
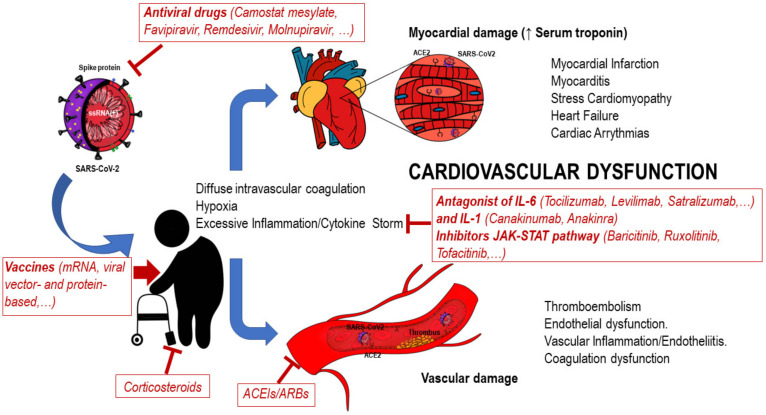
Impact of SARS-CoV-2 infection on cardiovascular system in older adults. Entry of SARS-CoV-2 into host cells (ciliated nasal epithelium; cardiomyocytes and endothelial cells) is mediated by angiotensin-converting enzyme 2 (ACE2) receptor. Main cardiovascular clinical alterations include myocardial injury, myocarditis, acute coronary syndrome, cardiac arrhythmia and thromboembolism. Elevation of D-dimers or prothrombin time, factors involved in blood clotting, are hallmarks of SARS-CoV-2 virus infection. Therapies against SARS-CoV-2 are indicated.

**Figure 2 jcm-12-00488-f002:**
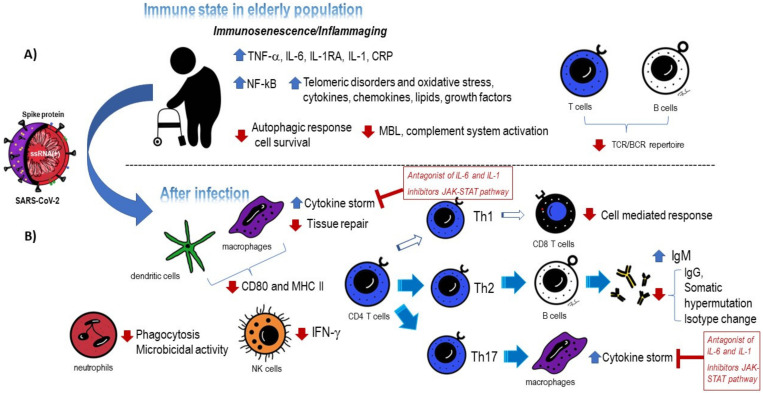
Immune state in elderly people and how SARS-CoV-2 infection affects immune system cells and soluble factors. (**A**) Several agents are involved in the physiological development of immunosenescence and inflammaging in elderly people. (**B**) A description of the different alterations that take place in elderly people after SARS-CoV-2 infection is shown. Antagonists of IL-6 and IL-1 and inhibitors of the JAK-STAT pathway are noted.

**Table 1 jcm-12-00488-t001:** Clinical trials underway to evaluate the efficacy and safety of intranasal vaccines or intranasal combined with intramuscular vaccines.

**Clinical Trial Identifier**	**Intervention**	**Status**	**Phase**
NCT04954287	CVXGA1 (IN)	Recruiting	I
NCT05522335	BBV154 (IN) and covaxin	Active, not recruiting	III
NCT04798001	MV-014-212 (IN)	Recruiting	I
NCT05248373	Gam-COVID-Vac (IN)	Not recruiting	I
NCT04751682	BBV154 (IN)	Completed	I
NCT04679909	AdCOVID (IN)	Active, not recruiting	I
NCT04871737	NDV (IN and IM)	Active, not recruiting	I
NCT05205746	NDV (IN)	Recruiting	II
NCT04839042	SC-Ad6-1(IN)	Recruiting	I
NCT05181709	NDV-HXP-S (IN and IM)	Recruiting	I

## Data Availability

Not applicable.

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
