# Peer review of "COVID-19: A Comprehensive Review on Cardiovascular Alterations, Immunity, and Therapeutics in Older Adults"

_jcm, 2023, doi:10.3390/jcm12020488_

Round 1

Reviewer 1 Report

The presented manuscript concerns the effects of cardiovascular and immune disorders on COVID-19, which is common in elderly patients.

The authors are correct to note that understanding the effects of COVID-19 on the cardiovascular and immune systems should translate into comprehensive and improved medical care for elderly patients with COVID-19, preventing cardiovascular and immunological alterations in this population.

1. However, poor reasoning does not allow the authors to identify and substantiate the peculiarities of the course of COVID-19 in elderly patients.

2. Innate response and inflammation in COVID-19 old adult population: the basic concepts of the immune system are presented.

3. Therapeutics against SARS-CoV-2 have out of the issue of the argued aim of the paper.

4. The clinical part is represented by a table with only 6 (six) clinical trials, which are not discussed in detail.

Reviewer 2 Report

The manuscript by José Rivera-Torres et al. entitled “Covid-19: A Comprehensive Review On Cardiovascular Alterations, Immunity And Therapeutics In The Older Adult Population” aimed to summarize the current knowledge supported by clinical findings both in the fields of cardiovascular and the immune system, and  to report current therapies.

I read with great interest this paper. The abstract summarizes the general significance of the manuscript and the article leads some evidence to such point; however, some major issues need to be addressed to improve the significance of the manuscript:

-        -  Cardiovascular (CV) involvement is a crucial complication in COVID-19, and no strategies are available to prevent or specifically address CV events in COVID patients. The identification of molecular partners contributing to CV manifestations in COVID-19 patients is crucial for providing early biomarkers, prognostic predictors and new therapeutic targets. Specifically, miRNAs have been proposed as valuable biomarkers and predictors of both cardiac and vascular damage occurring in SARS-CoV-2 infection. Consequently, this article could be discussed: Izzo C, Visco V, Gambardella J, et al. Cardiovascular implications of miRNAs in COVID-19 [published online ahead of print, 2022 Jul 2]. J Pharmacol Exp Ther. 2022;JPET-MR-2022-001210. doi:10.1124/jpet.122.001210.

 -         - “Cardiovascular alteration” section should be focused on the older adult population (as reported in the title).

-The “Conclusions” paragraph is too long.  The authors should summarize this section.

 -          It would be useful to add some figures, to make the article easier to read.

Round 2

Reviewer 1 Report

The presented manuscript discloses antivirals, cytokine antagonists, cytokine signaling pathway inhibitors, and vaccines to prevent cardiovascular and immunological alterations in elderly patients with COVID-19.

1. Despite the significance of the matter, the review is very brief and the paper format is more in accordance with Brief Communication than with Review.

2. The article lacks tables with the resume of clinical trials, case studies, and clinical findings.

3. The quantitative analysis is required, as well as the original schemes and diagrams.

4. The authors did not consider the influence of the patient's genetics on the severity of COVID-19:

- van der Made, C.I.; Netea, M.G.; van der Veerdonk, F.L.; Hoischen, A. Clinical implications of host genetic variation and susceptibility to severe or critical COVID-19. Genome Medicine, Genome Med 2022, 14, 96. https://doi.org/10.1186/s13073-022-01100-3

- Gozman, L.; Perry, K.; Nikogosov, D.; Klabukov, I.; Shevlyakov, A.; Baranova, A. A Role of Variance in Interferon Genes to Disease Severity in COVID-19 Patients. Front. Genet. 2021, 12, 709388. https://doi.org/10.3389/fgene.2021.709388
